

# The ΦBT1 large serine recombinase catalyzes DNA integration at pseudo-*attB* sites in the genus *Nocardia*

Marion Herisse[1], Jessica L. Porter[1], Romain Guerillot[1], Takehiro Tomita[2], Anders Goncalves Da Silva[1,2], Torsten Seemann[1,2], Benjamin P. Howden[1,2,3], Timothy P. Stinear[1,2,3] and Sacha J. Pidot[1]

[1] Department of Microbiology and Immunology at the Doherty Institute, University of Melbourne, Melbourne, VIC, Australia
[2] Microbiological Diagnostic Unit, University of Melbourne, Melbourne, VIC, Australia
[3] Doherty Applied Microbial Genomics, University of Melbourne, Melbourne, VIC, Australia

Corresponding authors
Timothy P. Stinear,
tstinear@unimelb.edu.au
Sacha J. Pidot,
spidot@unimelb.edu.au

## ABSTRACT

Plasmid vectors based on bacteriophage integrases are important tools in molecular microbiology for the introduction of foreign DNA, especially into bacterial species where other systems for genetic manipulation are limited. Site specific integrases catalyze recombination between phage and bacterial attachment sites (*attP* and *attB*, respectively) and the best studied integrases in the actinomycetes are the serine integrases from the *Streptomyces* bacteriophages ΦC31 and ΦBT1. As this reaction is unidirectional and highly stable, vectors containing phage integrase systems have been used in a number of genetic engineering applications. Plasmids bearing the ΦBT1 integrase have been used to introduce DNA into *Streptomyces* and *Amycolatopsis* strains; however, they have not been widely studied in other actinobacterial genera. Here, we show that vectors based on ΦBT1 integrase can stably integrate into the chromosomes of a range of *Nocardia* species, and that this integration occurs despite the absence of canonical *attB* sites in these genomes. Furthermore, we show that a ΦBT1 integrase-based vector can insert at multiple pseudo-*attB* sites within a single strain and we determine the sequence of a pseudo-*attB* motif. These data suggest that ΦBT1 integrase-based vectors can be used to readily and semi-randomly introduce foreign DNA into the genomes of a range of *Nocardia* species. However, the precise site of insertion will likely require empirical determination in each species to avoid unexpected off-target effects.

## INTRODUCTION

The actinomycetes comprise diverse bacterial genera, ranging from those well known for their prolific production of secondary metabolites (*Streptomyces* spp.) to those that contain some of world's most successful pathogens (*Mycobacterium* spp.). *Nocardia* species are actinomycetes that have features in common with pathogens and secondary metabolite producers. For example, several species of *Nocardia* (i.e., *Nocardia asteroides*, *Nocardia brasiliensis*) are opportunistic human and animal pathogens, causing lung, skin

and brain infections, predominantly in immunocompromised hosts, while others are reservoirs of bioactive natural products (*Hoshino et al., 2011*; *Tanaka et al., 1997*) and are capable of degrading or converting widespread environmental pollutants (*Hahn et al., 2013*; *Santillan et al., 2016*). These beneficial properties and their potential biotechnical applications have attracted a great deal of interest in the genus *Nocardia*. However, genetic tools to manipulate *Nocardia* spp. are limited in comparison to better studied actinomycetes, such as *Streptomyces* spp.

Serine integrases are widely used tools in the molecular biology of a range of species, especially actinomycetes. These integrase proteins (Int) are phage-encoded enzymes that mediate the site-specific recombination of DNA molecules between two sites (known as "attachment sites," or *att*-sites) on the phage (*attP*) and host genome (*attB*) (*Rutherford & Van Duyne, 2014*). As excision of the integrated DNA requires a separate protein (recombination directionality factor), the integration reaction is both highly directional and stable (*Fogg et al., 2014*). It is these two key features that have made these enzymes such attractive tools for the modification of bacterial and mammalian genomes (*Stark, 2017*).

Perhaps the best studied serine integrases are those derived from *Streptomyces* bacteriophages, including ΦC31 and ΦBT1. Vectors based on the integrase (*int*) from ΦC31 and ΦBT1 have been utilized to insert DNA into a range of genomes in both prokaryotes (including bacteria and archaea) and eukaryotes (yeast, plant and mammalian cells) (*Baltz, 2012*). The *attP* and *attB* sites for members of the serine recombinase family are often <50 bp in length and contain imperfect inverted repeat sequences that surround a core dinucleotide, known as the crossover site (*Smith & Thorpe, 2002*). In *Streptomyces coelicolor*, both ΦC31 Int and ΦBT1 Int mediate DNA insertion into separate genes (SCO3798, a pirin homologue and SCO4848, a putative membrane protein, respectively), inactivating their targets (*Gregory, Till & Smith, 2003*; *Kuhstoss, Richardson & Rao, 1991*). While ΦC31 *attB* sites are highly similar in a number of *Streptomyces* species, non-specific integration in sequences that have partial identity to *attB* (pseudo-*attB* sites) has also been reported, although this appears to occur relatively infrequently (*Baltz, 2012*; *Combes et al., 2002*). While vectors utilizing ΦBT1 Int have also been shown to integrate in a range of *Streptomyces* species and *A. mediterranei*, their usage outside these organisms has been limited (*Gregory, Till & Smith, 2003*; *Li et al., 2017*). Furthermore, the extent to which ΦBT1 Int can catalyze integration at pseudo-*attB* sites is unknown.

Here, we show that ΦBT1 Int can be used to integrate foreign DNA into a range of *Nocardia* species and that these insertions occur despite the absence of a canonical *attB* site. We show that these pseudo-*attB* sites are unique to each species, that certain sites are preferred within certain species and that minimal homology to the canonical *attB* is required for DNA insertion. Furthermore, using this range of unique pseudo-*attB* sites we identify a pseudo-*attB* motif and delineate the nucleotides important for DNA integration.

## METHODS

### Microbial culture

*Nocardia* strains were routinely cultured in Brain Heart Infusion (BHI) broth or on BHI agar plates (Difco). *E. coli* strains were cultured in Luria broth or on Luria agar plates.

*Streptomyces* species were cultured on MS agar for conjugation and isolation of transconjugants (*Kieser et al., 2000*). All strains used in this study are listed in Table S1. The plasmid pRT801, an *E. coli-Streptomyces* integrating vector containing the BT1 *int* gene and *attP* site, *aac3(iv)* (encoding apramycin resistance) and a *Streptomyces oriT* for conjugative transfer, was used in this study and has been previously published (*Gregory, Till & Smith, 2003*).

## DNA extraction procedures

Genomic DNA was extracted from *Nocardia* strains according to a modified DNA isolation method based on two previously published protocols (*Belisle & Sonnenberg, 1998*) (*Gonzalez-y-Merchand et al., 1996*). Briefly, a 10 ml culture of cells was grown in BHI supplemented with 2% glycine for four days at 30 °C with shaking at 200 rpm. Cells were pelleted and resuspended in 2 ml Tris-HCl pH 8.0 and mixed with an equal volume of 1:1 chloroform: methanol for 10 mins with gentle shaking. Cells were pelleted at 8,000×*g*, 10 min, then both the organic and aqueous phases were removed, leaving only the cells. Cells were dried to remove all traces of chloroform and methanol and were resuspended in 475 µl of TE buffer, followed by the addition of lysozyme at a final concentration of 5 mg/ml. Cells were incubated for 2 h at 37 °C and then pelleted at 8,000×*g*, 10 min and the supernatant removed. The cell pellet was then resuspended in 500 µl lysis buffer (6 M guanidine HCl, 10 mM EDTA, 1 mM β-mercaptoethanol, 2% Tween 20, 1% IGEPAL CA630, 1% Triton X-100) and the cells were frozen in a dry ice/ethanol bath, followed by thawing at 60 °C. This was repeated three times, then the cell suspension was extracted twice with chloroform. The supernatant was removed to a separate tube and the DNA was precipitated using ethanol. DNA was resuspended in 50 µl of 10 mM Tris-HCl and stored at −20 °C.

Plasmid DNA was extracted from *E. coli* DH10B cells containing pRT801 using a plasmid mini-prep kit (Favorgen Biotech, Ping-Tung, Taiwan).

## Transformation of *Nocardia* strains

*Nocardia* strains were transformed according to the method of *Ishikawa et al. (2006)* (*Ishikawa et al., 2006*). In brief, *Nocardia* strains were grown in 100 ml BHI containing 2% glycine for four days at 30 °C prior to transformation. Cells were harvested and washed twice with 50 ml ice-cold water. The cells were then resuspended in 500 µl of ice-cold 10% glycerol and 50 µl was used per transformation and transferred to a chilled electroporation cuvette (2 mm gap). Approximately ~1 µg of DNA was added and cells were pulsed at 2.4 kV using an electroporator (Electro Cell Porator 600; BTX Inc., Holliston, MA, USA). To each transformation, 900 µl of BHI broth was added and the cells were incubated for 2 h at 37 °C. *N. terpenica*, *N. uniformis* and *N. brasiliensis* transformations were plated onto BHI plates containing 50 µg/ml of apramycin, while *N. arthritidis* transformations were plated onto BHI plates containing 100 µg/ml apramycin. All plates were incubated for 2–3 days at 30 °C. For each *Nocardia* strain analyzed for pRT801 insertions sites, transformants were selected from independent transformation reactions to avoid detecting sibling colonies. As such, transformants were

picked from 14 independent transformation reactions for *N. terpenica*; six independent transformations for both *N. brasiliensis* and *N. uniformis*; and five independent transformations for *N. arthritidis*.

## Conjugation of plasmids into *Streptomyces* strains

DNA was transferred from *E. coli* ET12567 (pUZ8002) to *Streptomyces* species by conjugation according to a previously published protocol (*Kieser et al., 2000*).

## Confirmation of pRT801 integration in *Streptomyces* strains

Insertion of pRT801 in *S. coelicolor* and *S. lividans* was determined by PCR using the following primers: ScBT1-F-CCCAATACGAAGGAGACGAT; ScBT1-R-GCTCATTCACAACGACAACG. Insertion of pRT801 in *S. albus* was also determined by PCR, but due to differences upstream of the *attB* site in *S. albus*, a different forward primer (SaBT1-F-GGGGTGGGGTTCTTCTCAC) was used in conjunction with the ScBT1-R primer.

## DNA sequencing and analysis

Complete sequencing of the *N. terpenica* genome was performed with a combination of PacBio SMRT and Illumina sequence data. *N. terpenica* DNA was extracted as outlined above and prepared for sequencing on the PacBio RSII using the Template Prep Kit 1.0 (PacBio, Menlo Park, CA, USA). Genomic DNA was size selected after adapter ligation using a BluePippin system (Sage Biosciences, Beverly, MA, USA) with a 10 kb cutoff. Adapter-ligated, circularized DNA was loaded onto a single SMRT cell at 0.2 nM and sequence data were captured with a 6 h movie time. To complete the genome sequence of *N. terpenica*, PacBio sequencing data was assembled using HGAP3, as implemented in the SMRT Portal (PacBio, Menlo Park, CA, USA), resulting in a single contig. This contig was polished three times using Quiver (PacBio, Menlo Park, CA, USA) before final indel error correction with Illumina reads using Snippy v3.2 (https://github.com/tseemann/snippy).

For all Illumina sequencing, DNA libraries were created using the Nextera XT DNA preparation kit (Illumina, San Diego, CA, USA) and whole genome sequencing was performed on the NextSeq platform (Illumina, San Diego, CA, USA) with 2 × 150 bp paired-end chemistry. A sequencing depth of >50× was targeted for each sample. Samples with Illumina only data were assembled with SPAdes (v 3.10.1) (*Bankevich et al., 2012*) and annotated with Prokka v 1.12 (*Seemann, 2014*). Sequencing reads are available under BioProject ID PRJNA433999. Table S2 lists each read set deposited in SRA, along with species name and corresponding *attB* site, as listed in Fig. S1 and Table S3 of the manuscript. The sequence of pRT801 has been deposited in GenBank under accession number MH192349.

Insertion sites of pRT801 were manually inspected and identified using Geneious v9.1.6. Motif analysis was performed using MEME (www.memesuite.org, *Bailey & Elkan (1994)*). Rarefaction analysis was performed in R v3.3.2 (*R Development Core Team, 2016*) to estimate the total number of pseudo-*attB* sites present in the *N. terpenica* genome. To identify sites with homology to the predicted pseudo-*attB* motif in *Nocardia* genomes,

FIMO was used (http://meme-suite.org/tools/fimo, *Grant, Bailey & Noble (2011)*) with a threshold value of $1e^{-08}$.

## Construction of *rpl*-based phylogenetic tree

Sequences of *rpl* genes (*rplB, rplJ, rplM, rplS* and *rplT*) were extracted from 68 *Nocardia* genome sequences present in GenBank, as well as the four sequenced *Nocardia* genomes from this study, using the Perl script *gene-puller.pl* (https://github.com/kwongj/gene-puller). The *rpl* sequences were manually curated to a consistent length and then concatenated to create a single 2,352 bp pseudo-sequence for each strain and aligned using Clustal W 2.1 (*Larkin et al., 2007*). A maximum likelihood phylogenetic tree was inferred with IQ-Tree under a GTR+F+I+G4 nucleotide substitution model using the alignment of concatenated sequences mentioned above. Branch support for the phylogeny was assessed with 1,000 bootstrap replicates (*Hoang et al., 2018*; *Nguyen et al., 2015*).

# RESULTS

## Sequencing of *Nocardia terpenica* AUSMDU00012715

As part of a separate natural product discovery project we sought to assess the genetic tractability of a human pathogenic *Nocardia sp* (strain ID: AUSMDU00012715). To first establish the species identity of the strain and provide a solid reference for subsequent molecular investigations, we subjected this isolate to combined PacBio SMRT and Illumina sequencing to produce a single 9,306,871 bp circular chromosome with 8,241 predicted CDS and 70 tRNA regions (Fig. 1). No plasmids were detected. Subsequent phylogenetic analysis by comparing five *rpl* gene sequences from this *Nocardia* isolate and 68 other *Nocardia* species (including three others sequenced as part of this study; Tables S1 and S3) showed that this isolate was most closely related to a previously sequenced *N. terpenica* isolate (14 pairwise SNP differences among a total of 950 potential variable positions), suggesting that it was a member of the species *N. terpenica* (Fig. 1).

## pRT801 integrates into the *Nocardia terpenica* genome

To assess the potential of *N. terpenica* AUSMDU00012715to accept foreign DNA, we began by transforming this isolate using electroporation with a range of plasmids that have been successfully used in *Nocardia* and other Actinobacteria. These plasmids included a range of replicating as well as integrating vectors. The replicative plasmids pNV19 and pNV119 (*Chiba et al., 2007*) were competently hosted by *N. terpenica* AUSMDU00012715, as was the ΦBT1 Int expressing vector pRT801 (*Gregory, Till & Smith, 2003*). However, transformation with other integrating vectors, including those utilizing the ΦC31 and ΦL5 integrases failed to produce colonies on selective media.

Following the isolation of potential pRT801 transformants on apramycin-containing media, we screened colonies by PCR for the *aac(3)iv* gene to confirm the presence of pRT801. As a ΦBT1 integrase system has not previously been reported to function in *Nocardia* species, we were unsure where pRT801 may have inserted in the genome of *N. terpenica* transformants. We sequenced the genomes of three randomly selected clones to rapidly identify insertion sites in these transformants. Surprisingly, an analysis of these

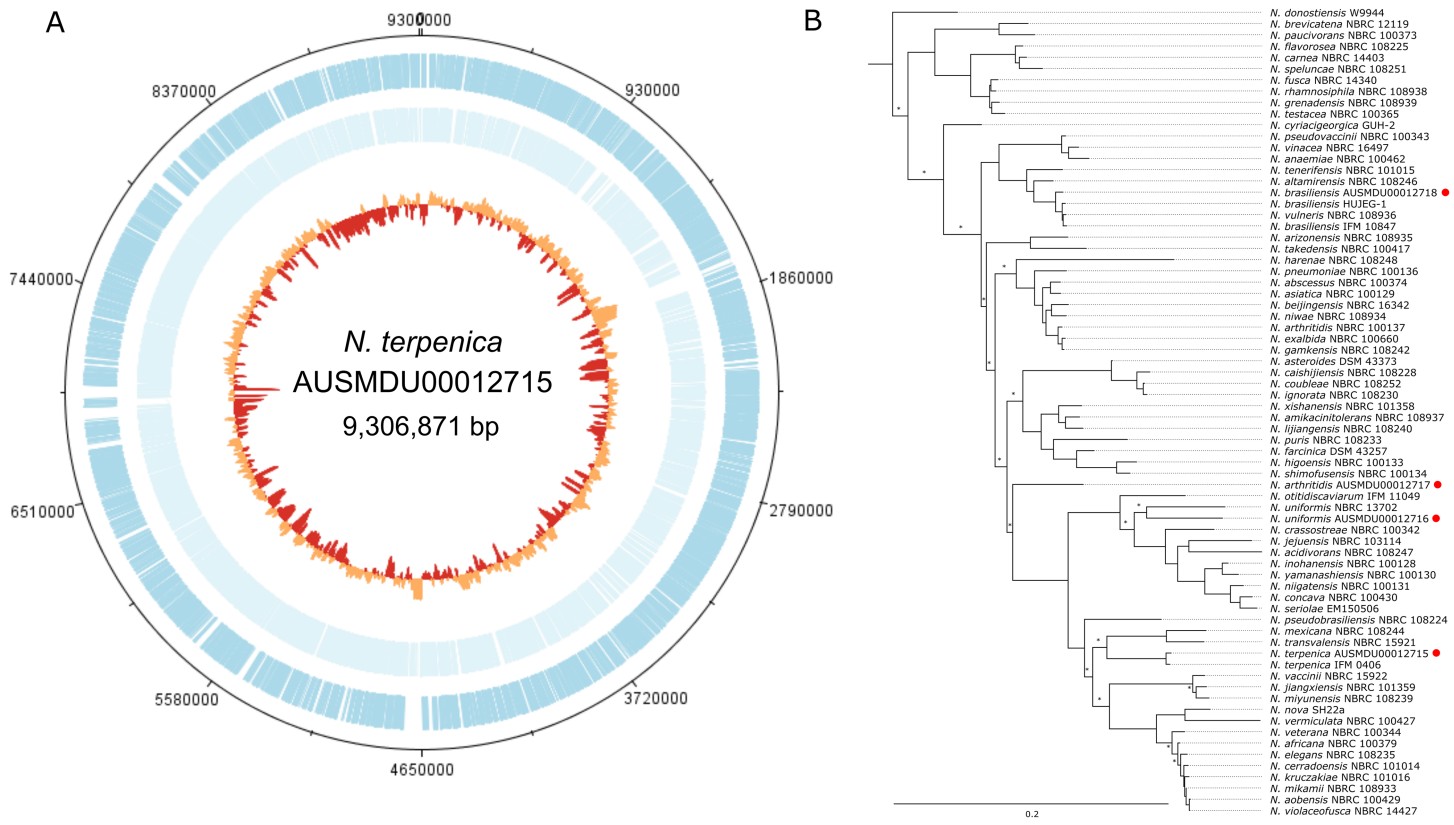

**Figure 1** **Genome and phylogenetic position of *Nocardia terpenica* AUSMD00012715.** (A) Genome map of *N. terpenica* 03-11436. Outer and inner blue and light blue lines represent CDS present on the forward and reverse strand, respectively. The innermost circle is a GC plot, showing variance in GC content across the genome. (B) Concatenated ribosomal gene based phylogenetic tree showing the position of *N. terpenica* 03-11436 among 71 other sequenced *Nocardia* strains (including other sequenced as part of this study, marked by red circles). A maximum likelihood phylogeny was inferred under a GTR+F+I+G4 model of nucleotide substitution, using as input an alignment of 2,352 bp concatenated nucleotide sequences for five ribosomal protein genes from 72 *Nocardia*. Asterisks indicate nodes with <70% bootstrap support (1,000 replicates).

genomes showed that pRT801 was inserted at a different chromosome location in each transformant, and not at a single unique *attB* site, as observed in *Streptomyces* species (*Gregory, Till & Smith, 2003*). Furthermore, these insertion sites were spread across the genome and had only limited homology to the canonical 73 bp *attB* site for ΦBT1 (*Gregory, Till & Smith, 2003*), the previously described minimal 36 bp *attB* site (*Zhang et al., 2008*), or even to the reported 9 bp common *attP-attB* sequence (*Gregory, Till & Smith, 2003*). An *in silico* scan of the *N. terpenica* genome based on these sequenced transformants did not find the canonical *S. coelicolor* ΦBT1 *attB* site within the genome, suggesting that all insertions in the *N. terpenica* genome were at pseudo-*attB* sites.

## All pRT801 insertion sites in *N. terpenica* occur at pseudo-*attB* sites

Although insertion into pseudo-*attB* sites has previously been observed in some actinomycetes following transformation with ΦC31 Int-based vectors (*Combes et al., 2002*; *Matsushima & Baltz, 1996*; *Murry et al., 2005*), the likelihood of insertion at these sites is up to 300 times lower than a true *attB* site (*Combes et al., 2002*). Furthermore,

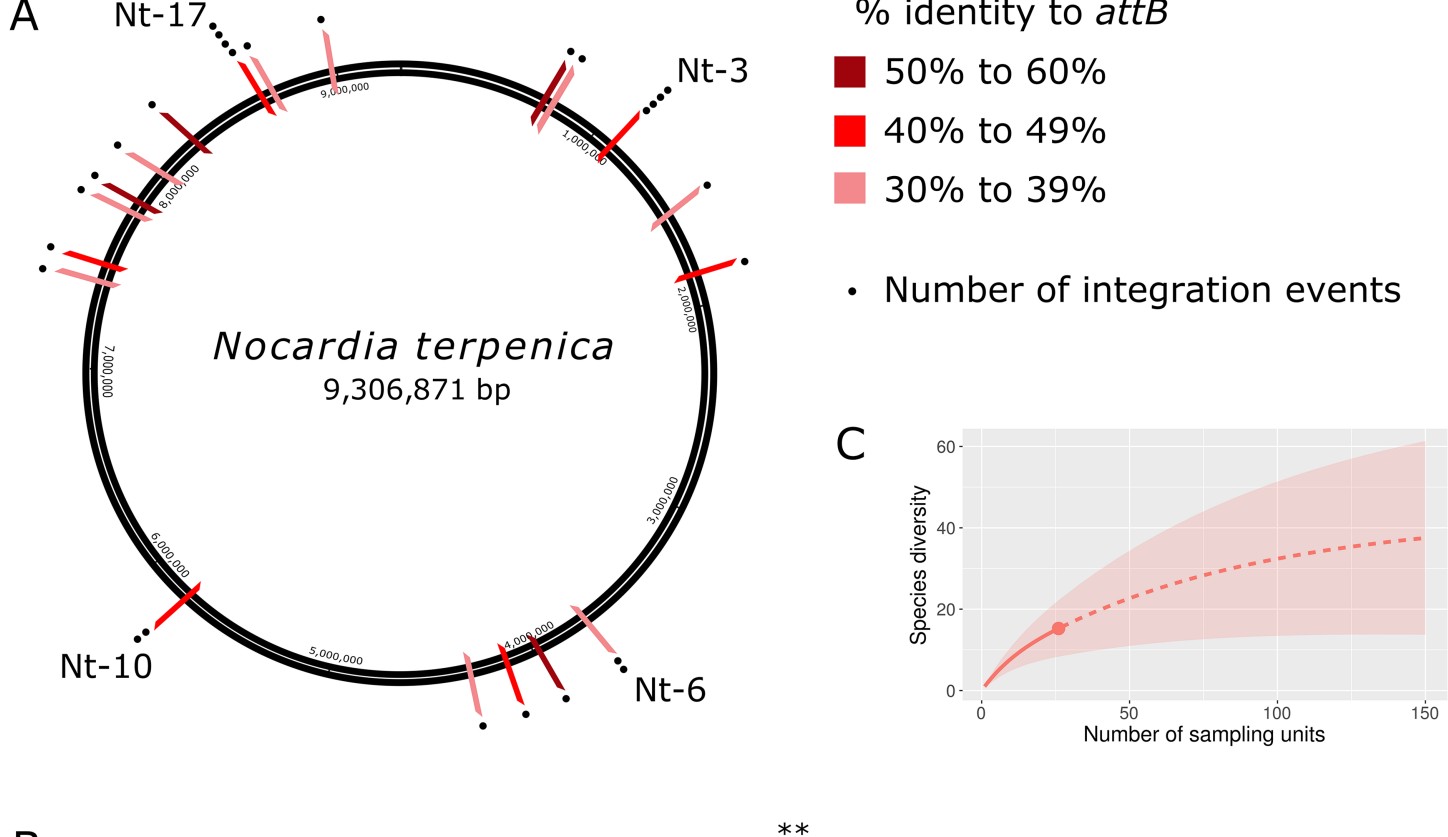

**Figure 2 Pseudo-*attB* insertion sites in *N. terpenica* AUSMD00012715.** (A) Locations of individual pRT801 insertion sites mapped onto the *N. terpenica* AUSMDU00012715 genome. Percentage similarity to the minimal *attB* site of *S. coelicolor* is indicated by varying shades of red. (B) Comparison of the most common *N. terpenica* pseudo-*attB* sites with the 73b canonical *attB* sequence from *S. coelicolor* (a comparison of all sites can be found in Fig. S1). Red nucleotides indicate identity to the canonical *attB* sequence. The common *attB–attP* core is boxed and the minimal *attB* sequence is underlined. The core GT dinucleotide is denoted by asterisks. (C) Rarefaction analysis showing number of predicted sites in the *N. terpenica* AUSMDU00012715 genome.

pseudo-*attB* sites have not been reported for ΦBT1 Int-based vectors. To characterize the extent and nature of these variant integration sites, we sequenced and analyzed the genomes of a further 24 *N. terpenica* pRT801 transformants. Analysis of this set of 27 transformants identified at least 19 individual pseudo-*attB* sites within the *N. terpenica* genome (Fig. 2A). Of the 27 identified pseudo-*attB* sites, two sites had four integration events each (Nt-3 and Nt-17), while a further two sites had two insertions each (Nt-6 and Nt-10) and the remaining 15 different sites had only a single integration event. The identified sites that had multiple integration events occurred in independent pRT801 transformations, meaning that these were not siblings, but rather that some pseudo-*attB* sites are preferred over others in the *N. terpenica* genome.

**Table 1 Analysis of pRT801 insertion sites in the *N. terpenica* genome and comparison with the known *attB* site.**

| Insert name | Identity to *attB* (%)[a] | Identity to minimal *attB* (%)[b] | Identity with *attB–attP* site[c] | Insertion events | Genome position | Encoded protein |
|---|---|---|---|---|---|---|
| Nt-1 | 26 (36%) | 19 (52%) | 5 | 1 | 731398 | MsrR family regulator |
| Nt-2 | 20 (27%) | 14 (38%) | 4 | 1 | 751814 | Disulfide bond forming protein |
| Nt-3 | 26 (36%) | 17 (47%) | 6 | 4 | 1066609 | Polyketide synthase |
| Nt-4 | 28 (38%) | 19 (52%) | 6 | 1 | 1505104 | Carboxymethylenebutenolidase |
| Nt-5 | 30 (41%) | 17 (47%) | 3 | 1 | 1826014 | Non-coding region |
| Nt-6 | 22 (30%) | 13 (36%) | 5 | 2 | 3715745 | Non-coding region |
| Nt-7 | 34 (46%) | 20 (55%) | 5 | 1 | 3939350 | FtsX-like permease |
| Nt-8 | 27 (37%) | 15 (41%) | 6 | 1 | 4126594 | Hypothetical protein |
| Nt-9 | 25 (34%) | 13 (36%) | 3 | 1 | 4286641 | Hypothetical protein |
| Nt-10 | 25 (34%) | 17 (47%) | 4 | 2 | 5761121 | Hypothetical protein |
| Nt-11 | 20 (27%) | 14 (38%) | 4 | 1 | 7436287 | FadD15 fatty acid ligase |
| Nt-12 | 26 (36%) | 16 (44%) | 3 | 1 | 7530197 | Hypothetical protein |
| Nt-13 | 25 (34%) | 13 (36%) | 3 | 1 | 7795264 | Putative deoxyribonuclease YcfH |
| Nt-14 | 25 (34%) | 19 (52%) | 5 | 1 | 7823861 | Non-coding region |
| Nt-15 | 25 (34%) | 12 (33%) | 4 | 1 | 8033744 | Telomeric repeat binding factor 2 |
| Nt-16 | 35 (48%) | 21 (58%) | 6 | 1 | 8246408 | MNT1/THI5-like protein |
| Nt-17 | 31 (42%) | 16 (44%) | 4 | 4 | 8620438 | Non-ribosomal peptide synthetase |
| Nt-18 | 22 (30%) | 14 (38%) | 5 | 1 | 8670073 | Hypothetical protein |
| Nt-19 | 23 (31%) | 14 (38%) | 4 | 1 | 8971553 | Phosphomevalonate kinase |

**Notes:**
[a] Number of identical nucleotide positions to the 73 bp canonical ΦBT1 *attB* site from *S. coelicolor*.
[b] Number of identical nucleotide positions to the 36 bp minimal ΦBT1 *attB* site from *S. coelicolor*.
[c] Number of identical nucleotide positions to the 9 bp *S. coelicolor attB*- ΦBT1 *attP* recombination site (*Gregory, Till & Smith, 2003*).

Analysis of the 27 insertion sites confirmed our initial observations and showed that these pseudo-*attB* sites were scattered across the chromosome (Fig. 2A), with insertions in a range of genes and non-coding regions (Table 1). Furthermore, we observed that even though recombination target site precision was 100%, with no deletions or insertions, these sites had only limited homology to the canonical ΦBT1 *attB* site (*Gregory, Till & Smith, 2003*) (Fig. 2B). Indeed, a comparison between all *N. terpenica* pRT801 insertion sites and the previously described ΦBT1 *attB* site showed that the only sequence that was completely conserved was a core GT dinucleotide (Fig. 2B and Fig. S1). A comparison of the *N. terpenica* pseudo-*attB* sites with the canonical 73 bp *attB* site revealed that nucleotide identity among these sites was between 27% and 48% (Table 1). Comparison with the minimal 36 bp *attB* site increased the percentage identity somewhat; however, it was still below 60% for all analyzed sites (Table 1). This suggests that, even though these pseudo-*attB* sites differ from the reported *S. coelicolor attB* by greater than 40% (a minimum of 14 out of 36 non-matching nucleotides), ΦBT1 Int is still capable of recognizing these sites and integrating donor DNA. We also noticed that the insertion sites that had more than one integration event were not the sites with the highest levels of similarity to the canonical 73 bp *attB*-site or even the 36 bp minimal *attB* site (Table 1). It appeared that greater nucleotide identity over the minimal *attB* region correlated with

greater identity over the whole *attB* sequence, but that this did not necessarily correlate with greater identity over the *attB-attP* site. These observations indicate that the "site-specific" nature of ΦBT1 integration events are not dependent on strict adherence to the canonical *attB* DNA sequence, but rather are tolerant to alterations within this sequence and suggest that DNA topological factors (spacing between certain nucleotides) may also be important.

Due to the diversity of the pseudo-*attB* sites identified in *N. terpenica* and the fact that some sites were identified more frequently than others, we sought to predict how many possible pseudo-*attB* sites were present in the genome. Rarefaction analysis was performed, which projected a total of 40 insertion sites in the *N. terpenica* genome (Fig. 2C), based on those already sequenced. This analysis suggests that there are many more potential insertion sites in the *N. terpenica* genome.

## Identification of pseudo-*attB* sites in other *Nocardia* species

To confirm whether this phenomenon of pseudo-*attB* site insertion was seen in other *Nocardia* species, we transformed pRT801 into three other clinical *Nocardia* isolates. As for *N. terpenica*, apramycin resistant colonies of *N. brasiliensis*, *N. arthritidis* and *N. uniformis* were obtained and the presence of *aac(3)iv* in each transformant was confirmed by PCR. The genomes of several transformants from each species were sequenced, which showed, similarly to *N. terpenica*, that pRT801 had inserted at pseudo-*attB* sites in all three organisms. Furthermore, none of the pseudo-*attB* insertion sites in *N. uniformis*, *N. arthritidis* or *N. brasiliensis* were the same as those seen in *N. terpenica* and none of the insertion sites were conserved across the different species.

Of the 12 sequenced *N. brasiliensis* AUSMDU00012716 isolates, nine different insertion sites were identified (Table S3), which ranged in identity to the canonical *attB* from 30% to 46% and from 33% to 52% when compared to the minimal *attB* site (Table S3). Likewise, a similar situation was seen for the nine sequenced *N. arthritidis* AUSMDU00012717 isolates, where five unique insertion sites were identified. Pseudo-*attB* sites found in this strain had between 34% and 42% identity to the canonical *attB* site, which increased to 41–58% when only the minimal *attB* site was investigated (Table S3). The range of identity to both the full-length *attB* and minimal *attB* from both of these strains is similar to that seen for *N. terpenica* insertion sites. The number of single insertion sites identified in these strains (Table S3; Fig. S1) suggests that further pseudo-*attB* sites remain to be identified. Again, as for *N. terpenica*, we did not find a correlation between sites with the highest identity to *attB* and an increased number of insertions. Interestingly, although the identity between the 9 bp *attB-attP* overlap site ranged from 4 to 6 bp, it appears that a minimum 2 bp overlap at this site (Nb-6 (Fig. S1), where only the essential GT dinucleotide was conserved) is all that is required for insertion.

Analysis of the 14 sequenced *N. uniformis* AUSMDU00012718 transformants showed a slightly different pattern, where only two unique insertion sites were identified. The preferred site, found in 13 of the 14 transformants, which we have labelled Nu-1, had 35 identical nucleotides (48%) to the canonical *attB* sequence and 22 nucleotides (61%)

that were identical to the minimal ΦBT1 *attB* sequence (Table S3; Fig. S1). The other pseudo-*attB* site in this species (named Nu-2) was more dissimilar to both the canonical and minimal *attB* sites than Nu-1 (28/73 (38%) nucleotides and 17/36 identical nucleotides (47%), respectively). It is possible that other pseudo-*attB* sites exist in this strain; however, their usage is likely to be rare. Nu-1 had the highest identity to *attB* seen in this study, perhaps explaining why Nu-1 appears to be the preferred insertion site in *N. uniformis*. However, given that the level of identity between Nu-1 and the minimal *attB* is still only 61% (14 of 36 non-identical nucleotides), this suggests that increasing identity of pseudo-*attB* with the canonical *attB* results in preferential insertion into these sites, although this is not absolute. As for *N. terpenica*, we observed that insertion at each target site occurred with 100% precision, and confirmed the formation of pseudo-*attL* and pseudo-*attR* sites at both the left and right hand ends of each insertion, indicating that these insertions occurred at genuine pseudo-*attB* sites and were not the result of DNA repair mechanisms (Fig. S2).

## Integration of pRT801 plasmid in the genome of *Streptomyces* species

Non-specific integration at pseudo-*attB* sites has previously been reported in *Streptomyces* species for ΦC31 Int (*Baltz, 2012*; *Combes et al., 2002*). Sequencing of our *Nocardia* transformants allowed us to investigate individual insertion sites without any *a priori* knowledge of their location. Previous studies using ΦBT1 Int in actinomycetes have determined insertion only by PCR around the known *attB* site (*Gregory, Till & Smith, 2003*; *Li et al., 2017*), meaning that insertions at pseudo-*attB* sites would have been classed as "non-integrants" and most likely ignored. We sought to determine whether ΦBT1 pseudo-*attB* sites exist in various *Streptomyces* species and if these sites are utilized in strains with canonical *attB* sites.

To do this, we first investigated the homology between *attB* sites from 22 *Streptomyces* genomes available in Genbank. We chose species that had previously been shown to be competent for the insertion of plasmid vectors based on ΦBT1 Int (*Baltz, 2012*; *Gregory, Till & Smith, 2003*). A comparison of these *attB* sites showed that they range from 74% to 100% identity with *S. coelicolor attB* (Table S4; Fig. S3). Variations in these sequences exist within the minimal *attB* region, as well as within the essential 9 bp core site, confirming our *Nocardia* findings in which it appears that only partial identity between *attP* and *attB* at this crossover site is sufficient for integration.

To investigate whether pseudo-*attB* insertion sites could be identified in *Streptomyces* species, we introduced pRT801 into *S. coelicolor*, *S. lividans*, *S. albus* and *S. cinnamonensis* by conjugation. As the genome sequences for *S. coelicolor*, *S. lividans* and *S. albus* are available, we designed PCR primers surrounding the *attB* insertion sites in these genomes, where a PCR product would confirm insertion of pRT801 at the canonical *attB* site. As the *S. albus* ΦBT1 *attB* site has 82% DNA identity to *attB* from *S. coelicolor*, we hypothesized that off-target insertions may be more likely in *S. albus*. A total of 50 *S. coelicolor*, *S. lividans* and *S. albus* transconjugant colonies were screened by PCR and
```
                        **
attB site (S. coelicolor)   GCCCGCTGCCGTCCTTGACCAGGTTTTTGACGAAAGTGATCCAGATGATCCAGCTCCACACCCCGAACGCGAG
S. coelicolor TS            GCCCGCTGCCGTCCTTGACCAGGTTTTTGACGAAAGTGATCCAGATGATCCAGCTCCACACCCCGAACGCGAG
S. cinnamonensis            GCCCGCTGCCGTCCTTCCACAGGTTCTTGACGAAAGTGATCCAGATGAACCAGCTCCACACCCCGAAGGCGAG
S. albus J1074              GCCCGCTCGCGTCGTTCCAGACATTCTTGACGAAGGTGACCCAGATGAACCAGCTCCACACCCCGAAGGCGAG
S. albus J1074 TS           GCCCGCTCGCGTCGTTCCAGACATTCTTGACGAAGGTGACCCAGATGAACCAGCTCCACACCCCGAAGGCGAG
```

**Figure 3** Comparison of *attB* sites in *Streptomyces* species. Comparison of *S. coelicolor* ΦBT1 *attB* with those from *S. albus* J1074 (obtained from GenBank), and *Streptomyces* strains sequenced as part of this study (*S. albus* J1074 (TS), *S. coelicolor* (TS) and *S. cinnamonensis*). The core GT site is in marked with asterisks, the minimal *attB* region is underlined and the 9 bp overlap region is marked with a box. Red nucleotides are those that differ from the canonical *attB* sequence.

all were positive for insertion of pRT801 into the canonical *attB* site (data not shown), indicating that if pseudo-*attB* site insertions do occur in these species, they are rare events. To confirm that integration had occurred in a single copy and that there were not alterations to the insertion site, we sequenced the genomes of a *S. coelicolor* and a *S. albus* transconjugant. As the genome sequence for *S. cinnamonensis* was unavailable, we initially sequenced four apramycin resistant transconjugants to identify the *attB* site in this species. Integration occurred at the same site in the four sequenced strains and alignment of the *S. cinnamonensis* ΦBT1 minimal *attB* with that from *S. coelicolor* showed 91% identity, with a 100% overlap in the 9 bp core sequence (Fig. 3). This suggests that while ΦBT1 Int can insert DNA at the same genomic region in these various *Streptomyces* strains, Int has a certain level of tolerance to mutations within *attB* and also suggests that sequence length and topological factors may be more important than nucleotide identity for DNA integration.

## Delineating the ΦBT1 pseudo-*attB* site motif

To determine if there was a conserved DNA signature to each of the pseudo-*attB* sites, motif analysis using MEME was performed. The 36 bp minimal *attB* regions from all unique sites of the pRT801-containing *Nocardia* transformants (36 sites total) were analyzed. The only motif found to be completely conserved across all sequences was the crossover site, or core GT dinucleotide (Fig. 4). However, a 34 bp motif, centered on the core GT dinucleotide, was identified and appears to contain a weakly conserved inverted repeat sequence (Fig. 4A). MEME analysis showed the importance of an almost invariant T at position six of the motif, as well as highly conserved nucleotides at positions 1, 5, 29, 30 and 34 of the motif (Fig. 4A). These conserved positions include TTC at positions 5–7, which is mirrored by GAA at positions 28–30 and both are the same distance from the core GT dinucleotide (11 bases distal on either side) (Fig. 4A). These positions lie within the putative zinc ribbon binding domain region of ΦBT1 *attB* (*Rutherford et al., 2013*) and a number of corresponding bases in *attP* can be matched within each half site (Fig. 4B), showing the bases sufficient for recombination between *attP* and pseudo-*attB* sites. When examining the presence of the conserved motif nucleotides within the *Streptomyces attB* and *Nocardia* pseudo-*attB* sites, there is partial conservation of the motif within every site (Fig. 4A and Fig. S4). The sequences containing the closest match to the pseudo-*attB* motif are those from the Streptomycetes, perhaps explaining why ΦBT1 integrase-based insertions at pseudo-*attB*

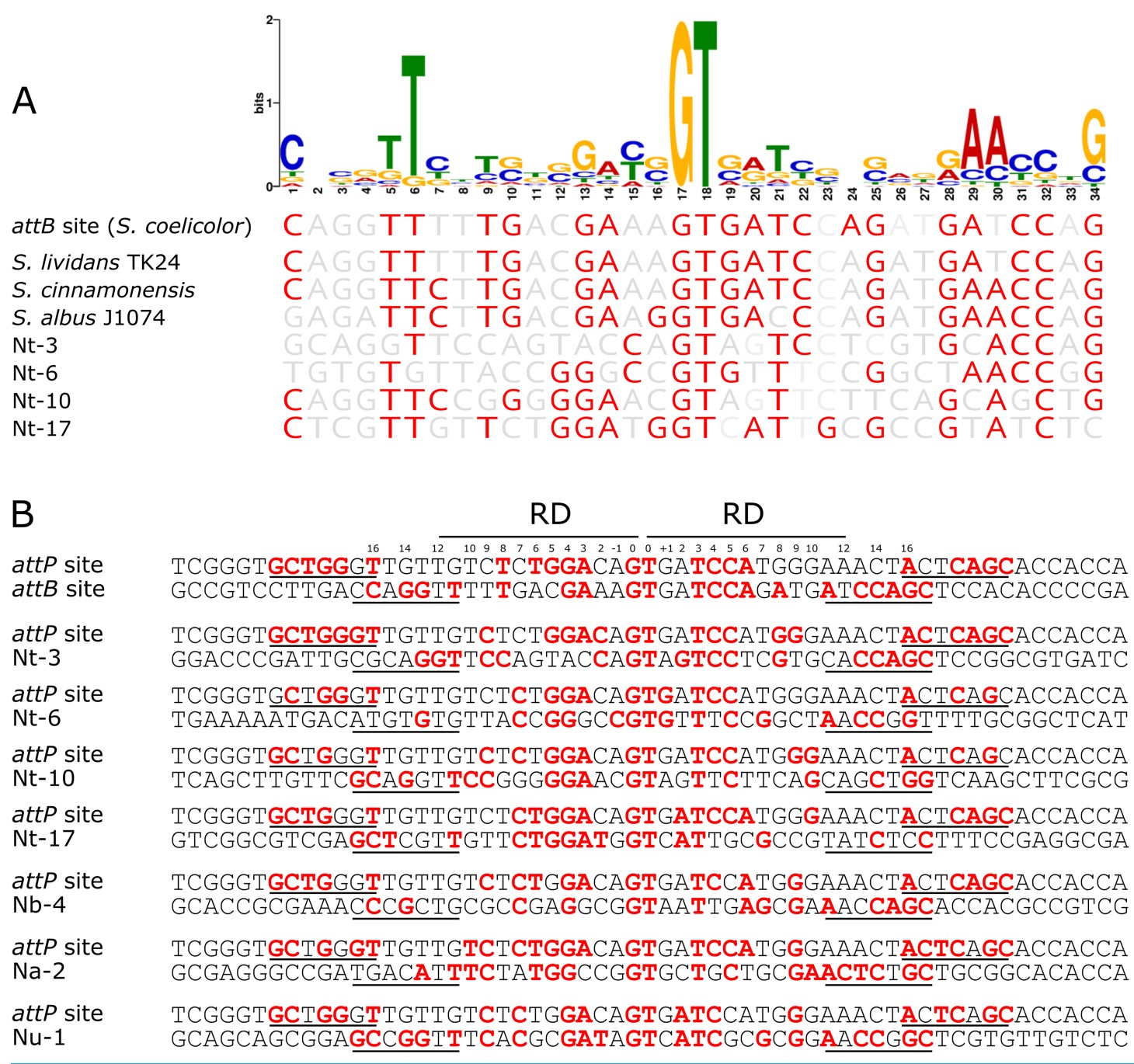

**Figure 4 The pseudo-*attB* motif in *Nocardia* species and comparison of pseudo-*attB* sites with *attP*.** (A) Alignment of *attB* sites with the pseudo-*attB* motif identified from *Nocardia* pseudo-*attB* insertion sites. Red nucleotides are those that are identical to the most common nucleotide at each position within the predicted motif. (B) Alignment of ΦBT1 *attP* and *attB* sites. The canonical *attP*–*attB* sites from ΦBT1 and *S. coelicolor*, respectively, are shown at the top, followed by an alignment of *attP* with the most common *N. terpenica* pseudo-*attB* sites, as well as those pseudo-*attB* sites that were most commonly seen from the other *Nocardia* species investigated in this study. The four predicted ZD binding motifs for ΦBT1 Int on each *attP* and pseudo-*attB* half site are underlined, with the predicted RD binding site shown at the top. Residues that are identical in three of the four ZD motifs within each alignment are shown in red. Residues that are identical at three out of four positions within –/+10 of the RD binding region are also shown in red. The central GT dinucleotide is in red and labelled as position 0.

sites are not seen in these strains. Furthermore, these data suggest that the presence of an imperfect inverted repeat of approximately the same length as the minimal *attB* sequence, surrounding a GT dinucleotide, is sufficient for ΦBT1 Int catalyzed DNA integration (Fig. 4).

To confirm our motif prediction, we performed a search for the pseudo-*attB* motif in the *N. terpenica* AUSMDU00012715 genome using FIMO (*Grant, Bailey & Noble, 2011*). A total of 52 sites with significant homology to the combined *Nocardia* pseudo-*attB* motif were identified at the relatively strict cut-off of $1e^{-08}$, which is in agreement with the potential number of sites estimated from our rarefaction analysis (Fig. 2C). A comparison of these sites identified by FIMO with those identified from the sequenced transformants found four matching sites. This suggests that there may still be additional refinements that can be made to the motif to further match previously identified pseudo-*attB* sites.

## DISCUSSION

As serine recombinase integration systems lead to unidirectional and highly stable integration of foreign DNA, these genetic tools have been widely used in both prokaryotes and eukaryotes for genetic engineering (*Stark, 2017*). ΦBT1 Int has been shown to integrate DNA site-specifically into a single site within *SCO4848* (encoding a membrane protein) in the *S. coelicolor* genome and the corresponding *attB*-site has been mapped (*Gregory, Till & Smith, 2003*). ΦBT1 Int has also been shown to integrate DNA into an orthologous gene in other *Streptomyces* species, such as *S. albus* and *S. lividans* (*Gregory, Till & Smith, 2003*). However, we have shown here that the nucleotide sequences of the *attB*-sites in these strains are not identical to the *S. coelicolor attB* sequence, revealing that the "site-specific" nature of insertion is tolerant to alterations at various positions within *attB*. Here we have shown that ΦBT1 Int can mediate DNA insertion into the genomes of several *Nocardia* species and that insertion occurs at pseudo-*attB* sites that have as little as 27% identity to the canonical *attB* site. Furthermore, our use of whole genome sequencing to rapidly identify and confirm the insertion site sequences meant that we were able to avoid the tedious digestion, relegation and selection of colonies required for plasmid rescue followed by Sanger sequencing of insertion sites. We propose that for similar studies looking at pseudo-insertion sites of integrases, sequencing costs are no longer prohibitive and allow for the use of whole genome sequencing to investigate such phenomena.

The integration sites for integrases from ΦC31 and ΦBT1 have been well characterized in *Streptomyces* species. Previous studies have shown that ΦC31 Int can catalyze DNA insertion at non-*attB* sequences in *S. coelicolor* in the absence of *attB* (*Combes et al., 2002*) and that pseudo-*attB* sites also exist in the human genome (*Thyagarajan et al., 2001*). However, pseudo-*attB* insertion sites for ΦBT1 Int have not been previously identified. Our data here provide the first description of ΦBT1 Int insertion of foreign DNA at pseudo-*attB* sites and show that insertions can occur at multiple pseudo-*attB* sites within each *Nocardia* genome. We have also shown that the essential requirement for insertion appears to be the core GT dinucleotide surrounded by an imperfect inverted repeat

sequence (Fig. 4). A previous study has shown that matching core dinucleotides between *attP* and *attB* of ΦBT1 are essential for recombination (*Zhang et al., 2010*). However, our data suggests that multiple substitutions within the minimal 36 bp *attB* site also do not impact on the ability for recombination to occur, suggesting that the requirements for integration in terms of sequence homology are relatively relaxed, provided the core GT dinucleotide is present.

A comparison of *attB* and *Nocardia* pseudo-*attB* sites revealed DNA identities in the range of 27–46% (20 to 34 nucleotides) (Table 1 and Table S3). This low level of identity (at most no better than one in two bases) is also reflected in identity with the minimal *attB* site, which is at similarly low levels. However, this lack of identity does not seem to inhibit integration, as some of these pseudo-*attB* sites were identified in multiple transformants from independent transformation reactions. Interestingly, it was not always the sites with the highest identity to *S. coelicolor attB* that were seen most often, but those integration sites that were seen most often do appear to have good conservation of bases that are involved in binding of the zinc ribbon domain (ZD) of ΦBT1 Int and of nucleotides that are putatively involved in binding of the recombinase domain (Fig. 4B).

Indeed, based on our deduced motif, certain nucleotides appear to be more important than others for DNA integration, which has also been noted with other large serine recombinases (*Rutherford & Van Duyne, 2014*). A previous study on the ΦC31 integrase showed that certain nucleotides within the minimal ΦC31-*attB* sequence are crucial for substrate recognition and interactions between the integrase and its attachment site (i.e., nucleotides −/+15 and −/+16), but also for the efficacy of recombination (i.e., nucleotides −/+2) (*Gupta, Till & Smith, 2007*). Interestingly, we found that across all the pseudo-*attB* sites, nucleotides at positions 1 and 34 (−/+16 from the core GT) are relatively well conserved, as are the nucleotides at positions 5 and 30 and 6 and 29 (−/+11 and −/+12, respectively) (Fig. 4A), which lie within the putative ZD binding region of ΦBT1 *attB* (*Rutherford et al., 2013*). Indeed, conservation of these sites aligns with the known mutation-sensitive bases present in the *attB* sites of other large serine recombinases, such as ΦC31 Int, which occur at similar locations from the core dinucleotide and appear to lie within the ZD and RD binding regions of these integrases (*Rutherford et al., 2013*; *Smith, 2015*). On the other hand, nucleotides at positions 11 and 24 (−/+6), 8 and 27 (−/+9) and 2 and 33 (−/+15) appear to have a random distribution and despite these sites being part of the putative RD and ZD binding regions, ΦBT1 Int binding appears to be relatively insensitive to substitutions at these positions. Furthermore, nucleotides at 16 and 19 (−/+1), and 14 and 21 (−/+3) are weakly conserved, which correlates well with previous data showing that mutations at these positions in the *S. coelicolor* ΦBT1 *attB* site do not affect integration (*Zhang et al., 2010*). However, a comparison of position −/+3 in *attP* and the most commonly observed pseudo-*attB* sites (Fig. 4B), shows that these positions correspond well to one another in each half site, suggesting that matching nucleotides at these positions may be part of the reason that these sites were preferred over others. As with the canonical *attB* sequence, the pseudo-*attB* motif identified here also exhibits a weakly conserved inverted repeat

centered on the core GT dinucleotide. As mentioned above, the most conserved bases appear between positions −/+11 to −/+16 from the core, representing approximately 1–1.5 helical turns of the DNA; which fits well with the *attP*-*attB*-Int binding model that has been previously proposed for phage A118 integrase (*Rutherford & Van Duyne, 2014*; *Rutherford et al., 2013*).

The identification of four sequenced pseudo-*attB* sites out of 52 predicted sites, using our motif as a search query, suggests that there are still additional refinements to be made to the motif. Indeed, as an equal number of mutants were not identified in each species, the contribution of sites from each to the motif is unequal, meaning that DNA compositional effects may have biased our motif sequence. It is possible that production of multiple motifs with each based on the pseudo-*attB* sequences obtained from an individual species may lead to more accurate prediction of putative pseudo-*attB* sites within each genome. Further work is required here to understand in detail the impact of changes to individual nucleotides within the ΦBT1 *attB* sequence.

Overall, we have shown that ΦBT1 integrase can be used as a tool to investigate the transformability of rare actinomycetes and that ΦBT1 Int-catalyzed DNA integration can occur at a number of pseudo sites. This study furthers our knowledge on the integration of serine recombinases and shows that the absence of a specific *attB*-site does not preclude integration of the phage. Given the potential number of diverse insertion sites in a non-*Streptomyces* genome, we suggest that caution should be used in interpreting the sites of phage integrase insertions in previously undescribed species.

## CONCLUSION

Plasmid vectors based on the ΦBT1 integrase have been used in multiple *Streptomyces* species and are important tools in these organisms. Here, we have shown that ΦBT1 Int-based vectors can be transformed into and can integrate stably in a range of *Nocardia* species. However, integration of pRT801 occurs at pseudo-*attB* sites in these organisms, and there are multiple pseudo-*attB* sites in each of these species. An in-depth analysis of these sites showed that only limited homology to the canonical *attB* sequence from *S. coelicolor* is required for insertion, although the core GT dinucleotide is essential. Furthermore, the delineation of a pseudo-*attB* motif has revealed those sites that are important for insertion and provides further information on the nucleotides that are important for recognition and integration by large serine recombinases.

### Funding

This work was supported by a National Health and Medical Research Council of Australia project grant (No. APP1105522). The funders had no role in study design, data collection and analysis, decision to publish, or preparation of the manuscript.

## Grant Disclosures

The following grant information was disclosed by the authors:
National Health and Medical Research Council of Australia project: APP1105522.

## Competing Interests

Timothy P Stinear is an Academic Editor for PeerJ.

## Author Contributions

- Marion Herisse conceived and designed the experiments, performed the experiments, analyzed the data, prepared figures and/or tables, authored or reviewed drafts of the paper, approved the final draft.
- Jessica L. Porter performed the experiments, authored or reviewed drafts of the paper, approved the final draft.
- Romain Guerillot analyzed the data, authored or reviewed drafts of the paper, approved the final draft.
- Takehiro Tomita performed the experiments, contributed reagents/materials/analysis tools, authored or reviewed drafts of the paper, approved the final draft.
- Anders Goncalves Da Silva analyzed the data, contributed reagents/materials/analysis tools, authored or reviewed drafts of the paper, approved the final draft.
- Torsten Seemann analyzed the data, contributed reagents/materials/analysis tools, authored or reviewed drafts of the paper, approved the final draft.
- Benjamin P. Howden analyzed the data, authored or reviewed drafts of the paper, approved the final draft.
- Timothy P. Stinear conceived and designed the experiments, analyzed the data, prepared figures and/or tables, authored or reviewed drafts of the paper, approved the final draft.
- Sacha J. Pidot conceived and designed the experiments, performed the experiments, analyzed the data, prepared figures and/or tables, authored or reviewed drafts of the paper, approved the final draft.

## DNA Deposition

The following information was supplied regarding the deposition of DNA sequences:

The sequencing data for all pRT801 mutants used for data analysis have been deposited in Genbank under BioProject PRJNA433999. The DNA sequence for plasmid pRT801 has been deposited in GenBank under accession number MH192349 and in the Supplemental Information.

## Data Availability

Raw sequencing data for all pRT801 mutants have been deposited in SRA under SRP133333.

## Supplemental Information

Supplemental information for this article can be found online at http://dx.doi.org/10.7717/peerj.4784#supplemental-information.

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
