# Peer review of "The ΦBT1 large serine recombinase catalyzes DNA integration at pseudo-attB sites in the genus Nocardia"

_PeerJ, doi:10.7717/peerj.4784_

## Round 0.1 · original submission · Minor Revisions

Your work received good comments from 3 independent reviewers. It is more a matter of providing further information in methods and editing some sections to improve the meaning of your claims. Please respond to each one of the reviewers and particularly pay attention to concerns raised by reviewers 1 & 2.

Reviewer 1 ·

Basic reporting

Generally, the manuscript is well written. The experiments are described clearly and the data presentation is fine. The sequences shown are relevant and sufficient. There are sections where the style of writing could be improved upon. These have been highlighted in the note to authors.

Experimental design

The research objective is clearly defined and the authors have chosen appropriate methods and analyses to address the question. The Methods section provides enough information to understand the work and evaluate the validity of the findings.

However, there are some incomplete information on the materials used. No details were provided on the source of the integrase and the integrating vector.

Validity of the findings

The findings from the work are valid and will be of interest to researchers using integrases for transgenesis and genome engineering studies. Similar analyses have been carried out for related integrases from ΦC31 and TG1, as well as ΦBT1 in Streptomyces. Results presented here will also make a direct contribution to studies on the reaction mechanism and sequence specificity of serine integrases.

Additional comments

In this report, Herisse et al. describe ΦBT1 integrase pseudo-attB sites in Nocardia sp. that recombined with canonical ΦBT1 attP site. The authors identified sequences with varying degrees of identity to published ΦBT1 attB sites, and attempted to explain the factors that influenced the accessibility of the integrase to these DNA sequences.
The findings are valid and would be of interest to researchers using integrases for transgenesis and genome engineering studies. Since, similar analyses have been carried out for related integrases from the phages ΦC31 and TG1, as well as ΦBT1 in Streptomyces, it will add to our understanding of the potential application of serine integrases. In addition, it could make a direct contribution to studies on the reaction mechanism and sequence specificity of serine integrases. I think it has the potential to attain the ‘publishable’ state, but it requires some careful revision.
The main area where improvement is required relates to a more effective use of relevant literature in putting the findings in context. There are some obvious inaccuracies reflecting inadequate familiarity with the relevant literature or weak understanding of the subject area. I have made a few specific comments below, which I believe will make the manuscript better, and more accessible to researchers working in the field.

The following needs to be attended to by the authors:
1. Line 21-23: The integrases do more than ‘bring attP and attB sites into contact’. Integrases are enzymes that catalyse recombination reactions between attP and attB sites, and this is not limited to ΦBT1 and ΦC31 integrases.
2. Line 54-55: The recombination directionality factor or RDF is not an enzyme in the strict sense. The integrase catalyses both the forward and reverse reactions, but specifically requires the RDF to activate the reverse reaction. There is no evidence that the RDF catalyses the reaction in the sense of acting as an enzyme.
3. Line 201:”homology to the canonical 73 bp attB”. What is this based on? Is there a reference to support it?
4. Line 212-213: “Furthermore, pseudo-attB sites have not been reported for ΦBT1 Int-based vectors”. This is not strictly true. Chen & Woo (Hum. Gene Ther. 2008; 19, 143-151) reported pseudo-attB sites for BT1 integrase in the human genome. The authors could include data from this paper in discussing their findings.
5. Line 272-275: It is not easily obvious to the reader what this sentence is implying. It will be clearer if the sequence of the attP site is included to help the reder see the 4-6 bp overlap alluded to in this sentence. See also Figure 3 legend. The authors may wish to include this in the supplementary information.
6. Line 294-295: Which previous studies are being referred to here? Again, including the relevant references will help put this point in context.
7. The authors have attempted to interpret the frequency of the pseudo sites and the base positions that are important. The data could probably be better explained by taking into considerations some detailed analysis reported in Rutherford et al (Nucleic Acids Res. 2013; 41, 8341-8356.) and Van Duyne & Rutherford (Curr. Opin. Struct. Biol. 2014; 24, 125-131.). In those two papers, the relationship between attB and attP sites were discussed extensively, and some of the principles there could greatly simplify the interpretation of data presented in this manuscript. Interestingly, neither of the two papers was cited here. See Lines 337-339, Lines 393-395, and Line 401-416.
8. Line 341-343 and Line 371-372: These sentences should be rewritten to make them clearer way. In its present state, it reads in an ambiguous way.
9. There are several examples where “ΦBT1 integrase” should be used rather than just “ΦBT1”. While “ΦBT1” is the phage, “ΦBT1 integrase” is the recombinase enzyme. See lines 28, 366, 377, 382, etc.. The same applies to ΦC31 and ΦC31 integrase (Line 405). The authors should adopt this nomenclature throughout the manuscript.
10. There are some incomplete information on the materials used. There are no details provided on the integrating vector, pRT801. What is it? Where is its source? In addition, what is the source of the integrases used? Were they obtained from another laboaratory? Or were they made by PCR? Gene synthesis? I could not find the relevant description in the supplementary information as well!

Reviewer 2 ·

Basic reporting

Strong. Please see points A, B, I, and J in the General Comments to the Authors section regarding wording and the reader's ability to interpret the raw data.

Experimental design

No comments other than those made in the in the General Comments to the Authors section.

Validity of the findings

The issue of controls is discussed in in the General Comments to the Authors section. I would like to see an additional figure with annotated sequences of the pseudo attL and attR sites showing the attP half-sites and pseudo attB half-sites.

Additional comments

I very much enjoyed reading this paper - it embodies a great deal of valuable work and has convinced me to revisit the debate as to whether or not pseudo sites for serine integrases exist.

The existence of pseudo sites for serine integrases has been the subject of some debate since they were first reported (1). Serine integrases bind attP and attB sites, bring them together, cleave them generating double strand breaks, the enzyme subunits rotate in relation to each other, and the attP and attB half-sites are ligated to make attL and attR sites. Much of the doubt around the existence of pseudo sites comes from the generation of the double strand breaks. How do we know that the integrase does not dissociate after the DNA is cleaved leaving endogenous DNA repair mechanisms like non-homologous end joining or double strand break repair to take over?

I have not yet found a report with controls sufficient to make me completely believe in the existence of pseudo sites (references [not limited to those] in reviews 2 and 3). Such controls would compare the frequency of integration into pseudo sites with frequency of integration into wild-type attB sites (frequency of integration = number of colonies plated on antibiotic / number of colonies plated on no antibiotic x 100); measure integration frequency in the absence of integrase; or use a strain in which DNA repair mechanisms were knocked out. Although I understand that inclusion of such controls would take considerable effort at this time, if the authors have any such data or data indicative of the frequency of pRT801 integration, I would strongly encourage its inclusion.

The work done to determine the sequence of a pseudo-attB motif in this paper is the most compelling evidence for the existence of pseudo att sites I have seen yet. What’s more, I believe there could be additional compelling evidence in the form of “pseudo attL and attR site” sequences from the multiple integration events observed in various pseudo attB sites. If these are the products of site-specific integration mediated by a serine integrase, the expected pseudo attL sequences would consist of the left arm (half-site) of the pseudo attB site and the right arm of the attP site joining at the central GT dinucleotide; and the pseudo attR sequences would consist of the right arm of the pseudo attB site and the left arm of the attP site joining at the central GT dinucleotide. I understand that this information would be available in the sequencing data provided and that reference to “no deletions or insertions” is made in Line 225, but if this information was presented in a figure, clearly annotating the pseudo attB half-sites and the attP half-sites within the pseudo attL and attR sites, this would provide a strong indication that integration events are not the products of DNA repair mechanisms. I believe this would be very valuable information within the field of site-specific recombination.

In addition to providing the sequences of pseudo attL and pseudo attR sites, please consider the following points:

A. Line 54: recombination directionality factors (as far as we know) don’t have enzymatic activity; it would be better to refer to them as proteins rather than enzymes.
B. Line 55 and throughout: Although much literature refers to recombination by serine integrases as “unidirectional” there is some evidence that the reaction can be reversed in the presence of integrase alone (4); “highly directional” may be a safer phrase.
C. Line 211-212: which reference refers to this likelihood?
D. Line 298: I’m not convinced frequency of insertion has been measured.
E. Line 314: positive for the presence of pRT801 or insertion of pRT801 into the genome?
F. Line 318: The reference to Figure 3 does not make sense here, please reconsider.
G. Line 351: Out of interest - of the 52 sites with significant homology to the combined Nocardia pseudo-attB motif found using FIMO, how many are in genes that would result in a lethal phenotype if pRT801 was inserted into them? Could it be that integration is happening at all of the 52 sites but 48 result in cell death?
H. Figure 2A: From reading the main text, should the key say “% identity to minimal attB site”?
I. After looking at the raw data deposited in SRA, a supplementary table explaining how read names (eg. gnl|SRA|SRR6764005.4.1) correspond to organisms would be helpful.
J. An accession number for the sequence of pRT801 would also be helpful.

Congratulations on your very interesting findings.

1. Thyagarajan et al. Site-Specific Genomic Integration in Mammalian Cells Mediated by Phage PhiC31 Integrase (2001) Molecular and Cellular Biology, 21;3926
2. Fogg et al. New Applications for Phage Integrases (2014) Journal of Molecular Biology, 426; 2703
3. Merrick et al. Serine Integrases: Advancing Synthetic Biology (2018) ACS Synthetic Biology, 16; 299
4. Zhang et al. Highly Efficient In Vitro Site-Specific Recombination System Based on Streptomyces Phage PhiBT1 Integrase (2008) Journal of Bacteriology, 190; 6392

Reviewer 3 ·

Basic reporting

This is a very nicely prepared manuscript. It is written in excellent English; literature is properly cited; and the figures are clear. I see no serious problems in these respects; minor comments below.

Experimental design

This is a straightforward study characterizing the ability of phiBT1 integrase to promote plasmid integration at genomic pseudosites in a range of bacterial species. The experimental work is to a high standard, and is properly described. The results are significant and clear. The work will be a useful contribution to knowledge in this area, and may be relevant especially to those who are investigating the molecular genetics of Nocardia, Streptomyces and other Gram-positive bacterial species.

Validity of the findings

I think the findings are valid and will be useful to others.

Additional comments

Overall I think that this is a clear, well presented article that will be a useful contribution to the literature. I have only a couple of minor points that I think should be addressed before publication.
1. Rarefaction analysis (Line 247, Figure 2C). I am not familiar with this analysis method. Some further details and/or a reference should be provided.
2. Line 345. English needs corrected.
3. Figure 1B. Text is very small - make sure it is all legible in the final version.
4. Figure 3. It might be better if the colour coding in this figure is the same as in Figures 2 and 4 (i.e. red indicates identity, not difference).

---

## Round 0.2 · accepted · Accept

I am thankful for your submission to PeerJ and appreciate your taking into account the suggestions provided by reviewers.

# Reviewer 1 ·

Basic reporting

This aspect has now been greatly improved and the issues raised in my initial review have been clarified to a satisfactory degree.

Experimental design

This aspect has now been greatly improved and the issues raised in my initial review have been clarified to a satisfactory degree.

Validity of the findings

This aspect has now been greatly improved and the issues raised in my initial review have been clarified to a satisfactory degree.

Additional comments

This aspect has now been greatly improved and the issues raised in my initial review have been clarified to a satisfactory degree.

Reviewer 2 ·

Basic reporting

No comment.

Experimental design

No comment.

Validity of the findings

No comment.

Additional comments

Thank you for taking the comments on board as far as possible. I am excited for the publication of this paper.